# S4ND: Modeling Images and Videos as Multidimensional Signals Using State Spaces

**Eric Nguyen**[*][†]**, Karan Goel**[*][‡]**, Albert Gu**[*][‡]**,**
**Gordon W. Downs**[‡]**, Preey Shah**[‡]**, Tri Dao**[‡]**, Stephen A. Baccus**[§]**, Christopher Ré**[‡]
[†]Department of BioEngineering, Stanford University
[‡]Department of Computer Science, Stanford University
[§]Department of Neurobiology, Stanford University
{etnguyen,albertgu,gwdowns,preey,trid,baccus}@stanford.edu
{kgoel,chrismre}@cs.stanford.edu

## Abstract

Visual data such as images and videos are typically modeled as discretizations of inherently continuous, multidimensional signals. Existing continuous-signal models attempt to exploit this fact by modeling the underlying signals of visual (e.g., image) data directly. However, these models have not yet been able to achieve competitive performance on practical vision tasks such as large-scale image and video classification. Building on a recent line of work on deep state space models (SSMs), we propose S4ND, a new multidimensional SSM layer that extends the continuous-signal modeling ability of SSMs to multidimensional data including images and videos. We show that S4ND can model large-scale visual data in 1D, 2D, and 3D as continuous multidimensional signals and demonstrates strong performance by simply swapping Conv2D and self-attention layers with S4ND layers in existing state-of-the-art models. On ImageNet-1k, S4ND exceeds the performance of a Vision Transformer baseline by $1.5\%$ when training with a 1D sequence of patches, and matches ConvNeXt when modeling images in 2D. For videos, S4ND improves on an inflated 3D ConvNeXt in activity classification on HMDB-51 by $4\%$. S4ND implicitly learns global, continuous convolutional kernels that are resolution invariant by construction, providing an inductive bias that enables generalization across multiple resolutions. By developing a simple bandlimiting modification to S4 to overcome aliasing, S4ND achieves strong zero-shot (unseen at training time) resolution performance, outperforming a baseline Conv2D by $40\%$ on CIFAR-10 when trained on $8 \times 8$ and tested on $32 \times 32$ images. When trained with progressive resizing, S4ND comes within $\sim 1\%$ of a high-resolution model while training $22\%$ faster.

## 1   Introduction

Modeling visual data such as images and videos is a canonical problem in deep learning. In the last few years, many modern deep learning backbones that achieve strong performance on benchmarks like ImageNet [53] have been proposed. These backbones are diverse, and include 1D sequence models such as the Vision Transformer (ViT) [13], which treats images as sequences of patches, and 2D and 3D models that use local convolutions over images and videos (ConvNets) [37, 25, 56, 58, 60, 42, 24, 32, 49, 64, 15].

A commonality among modern vision models capable of achieving state-of-the-art (SotA) performance is that they treat visual data as discrete pixels rather than continuous-signals. However, images

---

[*]Equal contribution.

36th Conference on Neural Information Processing Systems (NeurIPS 2022).

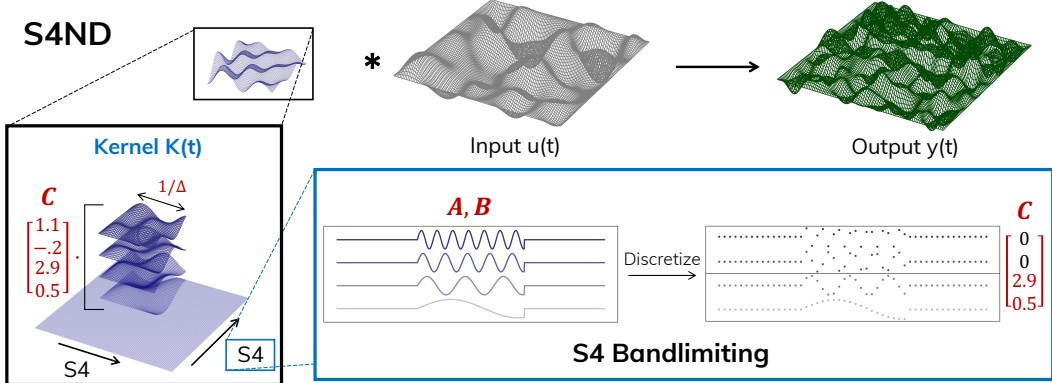

Figure 1: (**S4ND**.) (Parameters in *red*.) (*Top*) S4ND can be viewed as a depthwise convolution that maps a multidimensional input (*black*) to output (*green*) through a continuous convolution kernel (*blue*). (*Bottom Left*) The kernel can be interpreted as a linear combination (controlled by $C$) of basis functions (controlled by $A$, $B$) with flexible width (controlled by step size $\Delta$). For structured $C$, the kernel can further factored as a low-rank tensor product of 1D kernels, and can be interpreted as independent S4 transformations on each dimension. (*Bottom Right*) Choosing $A$, $B$ appropriately yields Fourier basis functions with controllable frequencies. To avoid aliasing in the final discrete kernels, the coefficients of $C$ corresponding to high frequencies can simply be masked out.

and videos are discretizations of multidimensional and naturally continuous signals, sampled at a fixed rate in the spatial and temporal dimensions. Ideally, we would want approaches that are capable of recognizing this distinction between data and signal, and directly model the underlying continuous-signals. This would give them capabilities like the ability to adapt the model to data sampled at different resolutions.

A natural approach to building such models is to parameterize and learn continuous convolutional kernels, which can then be sampled differently for data at different resolutions [16, 50, 54, 20, 21]. Among these, deep state space models (SSM) [20], in particular S4 [21], have achieved SotA results in modeling sequence data derived from continuous-signals, such as audio [17]. However, a key limitation of SSMs is that they were developed for 1D signals, and cannot directly be applied to visual data derived from multidimensional "ND" signals. Given that 1D SSMs outperform other continuous modeling solutions for sequence data [21], and have had preliminary success on image [21] and video classification [31], we hypothesize that they may be well suited to modeling visual data when appropriately generalized to the setting of multidimensional signals.

Our main contribution is S4ND, a new deep learning layer that extends S4 to multidimensional signals. The key idea is to turn the standard SSM (a 1D ODE) into a multidimensional PDE governed by an independent SSM per dimension. By adding additional structure to this ND SSM, we show that it is equivalent to an ND continuous convolution that can be factored into a separate 1D SSM convolution per dimension. This results in a model that is efficient and easy to implement, using the standard 1D S4 layer as a black box. Furthermore, it can be controlled by S4's parameterization, allowing it to model both long-range dependencies, or finite windows with a learnable window size that generalize conventional local convolutions [22].

We show that S4ND can be used as a drop-in replacement in strong modern vision architectures while matching or improving performance in 1D, 2D, and 3D. With minimal change to the training procedure, replacing the self-attention in ViT with S4-1D improves top-1 accuracy by $1.5\%$, and replacing the convolution layers in a 2D ConvNeXt backbone [42] with S4-2D preserves its performance on ImageNet-1k [10]. Simply inflating (temporally) this pretrained S4-2D-ConvNeXt backbone to 3D improves video activity classification results on HMDB-51 [38] by $4$ points over the pretrained 3D ConvNeXt baseline. Notably, we use S4ND as global kernels that span the entire input shape, which enable it to have global context (both spatially and temporally) in every layer of a network.

Additionally, we propose a low-pass bandlimiting modification to S4 that encourages the learned convolutional kernels to be smooth. While S4ND can be used at any resolution, performance suffers when moving between resolutions due to aliasing artifacts in the kernel, an issue also noted by prior work on continuous models [50]. While S4 was capable of transferring between different resolutions

on audio data [21], visual data presents a greater challenge due to the scale-invariant properties of images in space and time [52], as sampled images with more distant objects are more likely to contain power at frequencies above the Nyquist cutoff frequency. Motivated by this, we propose a simple criteria that masks out frequencies in the S4ND kernel that lie above the Nyquist cutoff frequency.

The continuous-signal modeling capabilities of S4ND open the door to new training recipes, such as the ability to train and test at different resolutions. On the standard CIFAR-10 [36] and Celeb-A [43] datasets, S4ND degrades by as little as $1.3\%$ when upsampling from low- to high-resolution data (e.g. $128 \times 128 \rightarrow 160 \times 160$), and can be used to facilitate progressive resizing to speed up training by $22\%$ with $\sim 1\%$ drop in final accuracy compared to training at the high resolution alone. We also validate that our new bandlimiting method is critical to these capabilities, with ablations showing absolute performance degradation of up to $20\%+$ without it.

## 2   Related Work

**Image Classification.** There is a long line of work in image classification, with much of the 2010s dominated by ConvNet backbones [37, 25, 56, 58, 60]. Recently, Transformer backbones, such as ViT [13], have achieved SotA performance on images using self-attention over a sequence of 1D patches [40, 39, 62, 71, 12]. Their scaling behavior in both model and dataset training size is believed to give them an inherent advantage over ConvNets [13], even with minimal inductive bias. Liu et al. [42] introduce ConvNeXt, which modernizes the standard ResNet architecture [25] using modern training techniques, matching the performance of Transformers on image classification. We select a backbone in the 1D and 2D settings, ViT and ConvNeXt, to convert into continuous-signal models by replacing the multi-headed self-attention layers in ViT and the standard Conv2D layers in ConvNeXt with S4ND layers, boosting or maintaining their top-1 accuracy on large-scale image classification.

**S4 & Video Classification.** To handle the long-range dependancies inherent in videos, [31] used 1D S4 for video classification on the Long-form Video Understanding dataset [67]. They first applied a Transformer to each frame to obtain a sequence of patch embeddings for each video frame independently, followed by a standard 1D S4 to model across the concatenated sequence of patches. This is akin to previous methods that learned spatial and temporal information separately [33], for example using ConvNets on single frames, followed by an LSTM [27] to aggregate temporal information. In contrast, modern video architectures such as 3D ConvNets and Transformers [24, 32, 49, 64, 15, 35, 41, 67, 2, 1] show stronger results when learning spatiotemporal features simultaneously, which the generalization of S4ND into multidimensions now enables us to do.

**Continuous-signal Models.** Visual data are discretizations of naturally continuous signals that possess extensive structure in the joint distribution of spatial frequencies, including the properties of scale and translation invariance. For example, an object in an image generates correlations between lower and higher frequencies that arises in part from phase alignment at edges [47]. As an object changes distances in the image, these correlations remain the same but the frequencies shift. This relationship can potentially be learned from a coarsely sampled image and then applied at higher frequency at higher resolution.

A number of continuous-signal models have been proposed for the visual domain to learn these inductive biases, and have led to additional desirable properties and capabilities. A classic example of continuous-signal driven processing is the fast Fourier transform, which is routinely used for filtering and data consistency in computational and medical imaging [11]. Neural Radiance Fields (NeRF) represents a static scene as a continuous function, allowing them to render scenes smoothly from multiple viewpoints [45]. CKConv [51] learns a continuous representation to create kernels of arbitrary size for several data types including images, with additional benefits such as the ability to handle irregularly sampled data. FlexConv [50] extends this work with a learned kernel size, and show that images can be trained at low resolution and tested at high resolution if the aliasing problem is addressed. S4 [21] increased abilities to model long-range dependancies using continuous kernels, allowing SSMs to achieve SotA on sequential CIFAR [36]. However, these methods including 1D S4 have been applied to relatively low dimensional data, e.g., time series, and small image datasets. S4ND is the first continuous-signal model applied to high dimensional visual data with the ability to maintain SotA performance on large-scale image and video classification.

**Progressive Resizing.** Training times for large-scale image classification can be quite long, a trend that is exacerbated by the emergence of foundation models [4]. A number of strategies have

emerged for reducing overall training time. Fix-Res [61] trains entirely at a lower resolution, and then fine-tunes at the higher test resolution to speed up training in a two-stage process. Mix-and-Match [28] randomly samples low and high resolutions during training in an interleaved manner. An effective method to reduce training time on images is to utilize progressive resizing. This involves training at a lower resolution and gradually upsampling in stages. For instance, fastai [14] utilized progressive resizing to train an ImageNet in under 4 hours. EfficientNetV2 [60] coupled resizing with a progressively regularization schedule, increasing the regularization as well to maintain accuracy. In EfficientNetV2 and other described approaches, the models eventually train on the final test resolution. As a continuous-signal model, we demonstrate that S4ND is naturally suited to progressive resizing, while being able to generalize to *unseen* resolutions at test time.

## 3 Preliminaries

**State space models.** S4 investigated state space models, which are linear time-invariant systems that map signals $u(t) \mapsto y(t)$ and can be represented either as a linear ODE (equation (1)) or convolution (equation (2)). Its parameters are $\boldsymbol{A} \in \mathbb{C}^{N \times N}$ and $\boldsymbol{B}, \boldsymbol{C} \in \mathbb{C}^N$ for a state size $N$.

$$\begin{aligned} x'(t) &= \boldsymbol{A}x(t) + \boldsymbol{B}u(t) \\ y(t) &= \boldsymbol{C}x(t) \end{aligned} \qquad (1) \qquad \begin{aligned} K(t) &= \boldsymbol{C}e^{t\boldsymbol{A}}\boldsymbol{B} \\ y(t) &= (K * u)(t) \end{aligned} \qquad (2)$$

**Basis functions.** For the clearest intuition, we think of the convolution kernel as a linear combination (controlled by $\boldsymbol{C}$) of **basis kernels** $K_n(t)$ (controlled by $\boldsymbol{A}, \boldsymbol{B}$)

$$K(t) = \sum_{k=0}^{N-1} \boldsymbol{C}_k K_k(t) \qquad K_n(t) = (e^{t\boldsymbol{A}}\boldsymbol{B})_n \qquad (3)$$

**Discretization.** The SSM (1) is defined over a continuous-time axis and produces continuous-time convolution kernels (2)(3). Given a discrete input sequence $u_0, u_1, \ldots$ sampled uniformly from an underlying signal $u(t)$ at a step size $\Delta$ (i.e. $u_k = u(k\Delta)$), the kernel can be sampled to match the rate of the input. Note that instead of directly sampling the kernel, standard discretization rules should be applied to minimize the error from the discrete to the continuous-time kernel [21]. For inputs given at different resolutions, the model can then simply change its $\Delta$ value to compute the kernel at different resolutions.

We note that the step size $\Delta$ does not have to be exactly equal to a "true sampling rate" of the underlying signal, but only the relative rate matters. Concretely, the discrete-time kernel depends only on the *product* $\Delta \boldsymbol{A}$ and $\Delta \boldsymbol{B}$, and S4 learns separate parameters $\Delta, \boldsymbol{A}, \boldsymbol{B}$.

**S4.** S4 is a special SSM with prescribed $(\boldsymbol{A}, \boldsymbol{B})$ matrices that define well-behaved basis functions, and an algorithm that allows the convolution kernel to be computed efficiently. Variants of S4 exist that define different basis functions, such as simple diagonal SSMs [23], or one that defines **truncated Fourier functions** $K_n(t) = \sin(2\pi nt)\mathbb{I}([0, 1])$ [22] (Fig. 1). These versions of S4 have easy-to-interpret basis functions that will allow us to control the frequencies in the kernel (Section 4.2).

## 4 Method

We describe the proposed S4ND model for the 2D case only, for ease of notation and presentation. The results extend readily to general dimensions; full statements and proofs for the general case are in Appendix A. Section 4.1 describes the multidimensional S4ND layer, and Section 4.2 describes our simple modification to restrict frequencies in the kernels. Fig. 1 illustrates the complete S4ND layer.

### 4.1 S4ND

We begin by generalizing the (linear time-invariant) SSM (1) to higher dimensions. Notationally, we denote the individual time axes with superscripts in parentheses. Let $u = u(t^{(1)}, t^{(2)})$ and $y = y(t^{(1)}, t^{(2)})$ be the input and output which are signals $\mathbb{R}^2 \rightarrow \mathbb{C}$, and $x =$

$(x^{(1)}(t^{(1)}, t^{(2)}), x^{(2)}(t^{(1)}, t^{(2)})) \in \mathbb{C}^{N^{(1)} \times N^{(2)}}$ be the SSM state of dimension $N^{(1)} \times N^{(2)}$, where $x^{(\tau)} : \mathbb{R}^2 \to \mathbb{C}^{N^{(\tau)}}$.

**Definition 1** (Multidimensional SSM). *Given parameters $\boldsymbol{A}^{(\tau)} \in \mathbb{C}^{N^{(\tau)} \times N^{(\tau)}}$, $\boldsymbol{B}^{(\tau)} \in \mathbb{C}^{N^{(\tau)} \times 1}$, $\boldsymbol{C} \in \mathbb{C}^{N^{(1)} \times N^{(2)}}$, the 2D SSM is the map $u \mapsto y$ defined by the linear PDE with initial condition $x(0, 0) = 0$ :*

$$\frac{\partial}{\partial t^{(1)}} x(t^{(1)}, t^{(2)}) = (\boldsymbol{A}^{(1)} x^{(1)}(t^{(1)}, t^{(2)}), x^{(2)}(t^{(1)}, t^{(2)})) + \boldsymbol{B}^{(1)} u(t^{(1)}, t^{(2)})$$

$$\frac{\partial}{\partial t^{(2)}} x(t^{(1)}, t^{(2)}) = (x^{(1)}(t^{(1)}, t^{(2)}), \boldsymbol{A}^{(2)} x^{(2)}(t^{(1)}, t^{(2)})) + \boldsymbol{B}^{(2)} u(t^{(1)}, t^{(2)}) \qquad (4)$$

$$y(t^{(1)}, t^{(2)}) = \langle \boldsymbol{C}, x(t^{(1)}, t^{(2)}) \rangle$$

Note that Definition 1 differs from the usual notion of multidimensional SSM, which is simply a map from $u(t) \in \mathbb{C}^n \mapsto y(t) \in \mathbb{C}^m$ for higher-dimensional $n, m > 1$ but still with 1 time axis. However, Definition 1 is a map from $u(t_1, t_2) \in \mathbb{C}^1 \mapsto y(t_1, t_2) \in \mathbb{C}^1$ for *scalar* input/outputs but over *multiple* time axes. When thinking of the input $u(t^{(1)}, t^{(2)})$ as a function over a 2D grid, Definition 1 can be thought of as a simple linear PDE that just runs a standard 1D SSM over each axis independently.

Analogous to equation (2), the 2D SSM can also be viewed as a multidimensional convolution.

**Theorem 1.** (4) *is a time-invariant system that is equivalent to a 2D convolution $y = K * u$ by the kernel*

$$K(t^{(1)}, t^{(2)}) = \langle \boldsymbol{C}, (e^{t^{(1)} \boldsymbol{A}^{(1)}} \boldsymbol{B}^{(1)}) \otimes (e^{t^{(2)} \boldsymbol{A}^{(2)}} \boldsymbol{B}^{(2)}) \rangle \qquad (5)$$

*This kernel is a linear combination of the $N^{(1)} \times N^{(2)}$ basis kernels $\{K_{n^{(1)}}^{(1)}(t^{(1)}) \otimes K_{n^{(2)}}^{(1)}(t^{(2)}) : n^{(1)} \in [N^{(1)}], n^{(2)} \in [N^{(2)}]\}$ where $K^{(\tau)}$ are the standard 1D SSM kernels (3) for each axis .*

However, a limitation of this general form is that the number of basis functions $N^{(1)} \times N^{(2)} \times \ldots$ grows exponentially in the dimension, increasing the parameter count (of $\boldsymbol{C}$) and overall computation dramatically. This can be mitigated by factoring $\boldsymbol{C}$ as a low-rank tensor.

**Corollary 4.1.** *Suppose that $\boldsymbol{C} \in \mathbb{C}^{N^{(1)} \times N^{(2)}}$ is a low-rank tensor $\boldsymbol{C} = \sum_{i=1}^r \boldsymbol{C}_i^{(1)} \otimes \boldsymbol{C}_i^{(2)}$ where each $\boldsymbol{C}_i^{(\tau)} \in \mathbb{C}^{N^{(\tau)}}$. Then the kernel (5) also factors as a tensor product of 1D kernels*

$$K(t^{(1)}, t^{(2)}) = \sum_{i=1}^r K_i^{(1)}(t^{(1)}) \otimes K_i^{(2)}(t^{(2)}) := \sum_{i=1}^r (\boldsymbol{C}_i^{(1)} e^{t^{(2)} \boldsymbol{A}^{(1)}} \boldsymbol{B}^{(1)}) \otimes (\boldsymbol{C}_i^{(2)} e^{t^{(2)} \boldsymbol{A}^{(2)}} \boldsymbol{B}^{(2)})$$

In our experiments, we choose $\boldsymbol{C}$ as a rank-1 tensor, but the rank can be freely adjusted to tradeoff parameters and computation for expressivity. Using the equivalence between (1) and (2), Corollary 4.1 also has the simple interpretion as defining an independent 1D SSM along each axis of the multidimensional input.

## 4.2 Resolution Change and Bandlimiting

SSMs in 1D have shown strong performance in the audio domain, and can nearly preserve full accuracy when tested zero-shot on inputs sampled at very different frequencies [21]. This capability relies simply on scaling $\Delta$ by the relative change in frequencies (i.e., if the input resolution is doubled, halve the SSM's $\Delta$ parameter). However, sampling rates in the spatial domain are often much lower than temporally, leading to potential aliasing when changing resolutions. A standard technique to avoid aliasing is to apply a low-pass filter to remove frequencies above the Nyquist cutoff frequency.

For example, when $\boldsymbol{A}$ is diagonal with $n$-th element $\boldsymbol{a}_n$, each basis function has simple form $K_n(t) = e^{t\boldsymbol{a}_n} \boldsymbol{B}_n$. Note that the frequencies are mainly controlled by the imaginary part of $\boldsymbol{a}_n$. We propose the following simple method: for any $n$ such that $\boldsymbol{a}_n \cdot \Delta < \frac{1}{2}\alpha$, mask out the corresponding coefficient of the linear combination $\boldsymbol{C}_n$ (equation (3)). Here $\alpha$ is a hyperparameter that controls the cutoff; theoretically, $\alpha = 1.0$ corresponds to the Nyquist cutoff if the basis functions are pure sinusoids. However, due to the decay $e^{\Re(\boldsymbol{a}_n)}$ arising from the real part as well as approximations arising from using finite-state SSMs, $\alpha$ often has to be set lower empirically.

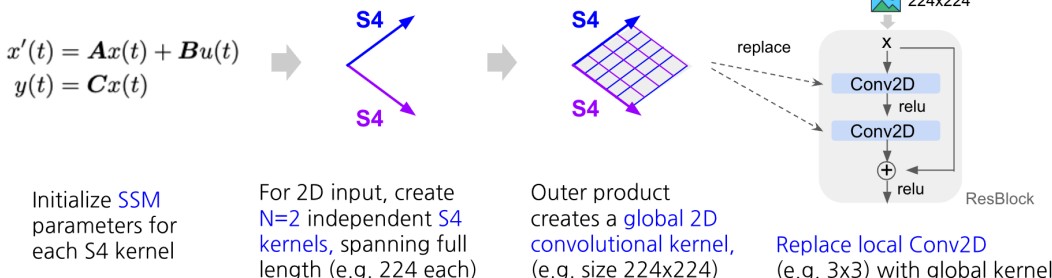

Figure 2: (**Flowchart of S4ND for images: 2D example.**) S4ND can process images as 2D inputs by initializing an SSM per spatial dimension $x$ and $y$ of the input. Two independent S4 kernels are then instantiated that span the entire input lengths of each dimension (e.g., 224 as shown above). Computing an outer product of the two 1D kernels produces a global convolutional kernel (e.g., 224x224). This global kernel can replace standard local Conv2D layers where ever they are used, such as ResNet or ConvNeXt blocks. A similar procedure can be done in 3D (with 3 S4 kernels) to create 3D global kernels for videos.

## 5 Experiments

We evaluate S4ND on large-scale image classification in Section 5.1 in the 1D and 2D settings, followed by activity classification in videos in Section 5.2 in the 3D setting, where using S4ND as a drop-in replacement for standard deep learning layers matches or improves performance in all settings. In Section 5.3, we performed controlled ablations to highlight the benefits of S4ND as a continuous-signal model in images.

### 5.1 S4ND in 1D & 2D: Large-scale Image Classification

First, we show that S4ND is a drop-in replacement for existing visual modeling layers such as 1D self-attention and 2D local convolutions, with no degradation in top-1 performance when used in modern backbones such as ViT [13] and ConvNeXt [42] on ImageNet-1k [10].

**Baselines and Methodology.** We consider large-scale image classification on the ImageNet-1k dataset, which consists of 1000 classes and 1.3M images. We start with two strong baselines: ViT-B (base, 88M parameters) for processing images in the 1D setting and ConvNeXt-T (tiny, 28.4M) in the 2D setting. (We omit the postfix "B" and "T" for brevity). More recent works using Transformers on images have surpassed ViT, but we focus on the original ViT model to highlight specifically the drop-in capability and performance difference in self-attention vs. S4ND layers. We first swap the self-attention layers in ViT with S4ND layers, and call this model S4ND-ViT. Notably, we simplify ViT by removing the positional encodings, as S4ND does not

Table 1: (**Performance on image classification.**) Top-1 test accuracy benchmarks for images in the 1D and 2D settings. ConvNeXt-M, for "micro", is a reduced model size for Celeb-A, while "-ISO" is an isotropic S4ND backbone [21].

| MODEL | DATASET | PARAMS | ACC |
|---|---|---|---|
| ViT-B | ImageNet | 88.0M | 78.9 |
| S4ND-ViT-B | ImageNet | 88.8M | **80.4** |
| ConvNeXt-T | ImageNet | 28.4M | 82.1 |
| S4ND-ConvNeXt-T | ImageNet | 30.0M | **82.2** |
| Conv2D-ISO | CIFAR-10 | 2.2M | 93.7 |
| S4ND-ISO | CIFAR-10 | 5.3M | **94.1** |
| ConvNeXt-M | Celeb-A | 9.2M | 91.0 |
| S4ND-ConvNeXt-M | Celeb-A | 9.6M | **91.3** |

require injecting this inductive bias. Similarly, we swap the local Conv2D layers in the ConvNeXt blocks with S4ND layers, which we call S4ND-ConvNeXt, a model with global context at each layer. Both S4ND variants result in similar parameter counts compared to their baseline models.

**Training.** For all ImageNet models, we train from scratch with no outside data and adopt the training procedure from [62, 69], which uses the AdamW optimizer [44] for 300 epochs, cosine decay learning rate, weight decay 0.05, and aggressive data augmentations including RandAugment [7], Mixup [72], and AugMix [26]. We add RepeatAug [29] for ConvNeXt and S4ND-ConvNeXt. The

initial learning rate for ViT (and S4ND-ViT) is 0.001, while for ConvNeXt (and S4ND-ConvNeXt) it is 0.004. See Appendix B.1 for additional training procedure details.

**Results.** Table 1 shows top-1 accuracy results for each model on ImageNet. After reproducing the baselines, S4ND-ViT was able to moderately boost performance by +1.5% over ViT, while S4ND-ConvNeXt matched the original ConvNeXt's performance. This indicates that S4ND is a strong primitive that can replace self-attention and standard 2D convolutions in practical image settings with large-scale data.

## 5.2 S4ND in 3D: Video Classification

Next, we demonstrate the flexible capabilities of S4ND in settings involving pretraining and even higher-dimensional signals. We use the activity recognition dataset HMDB-51 [38] which involves classifying videos in 51 activity classes.

**Baselines and Methodology.** Prior work demonstrated that 2D CNNs (e.g. pretrained on ImageNet) can be adapted to 3D models by 2D to 3D kernel *inflation* (I3D [5]), in which the 2D kernels are repeated temporally $N$ times and normalized by $1/N$. Our baseline, which we call ConvNeXt-I3D, uses the 2D ConvNeXt pretrained on ImageNet (Section 5.1) with I3D inflation. Notably, utilizing S4ND in 3D enables global context *temporally* as well. We additionally test the more modern spatial-temporal separated 3D convolution used by S3D [68] and R(2+1)D [65], which factor the 3D convolution kernel as the outer product of a 2D (spatial) by 1D (temporal) kernel. Because of its flexible factored form (Section 4.1), S4ND automatically has these inflation capabilities. We inflate the pretrained S4ND-ConvNeXt simply by loading the pretrained 2D model weights for the spatial dimensions, and initializing the temporal kernel parameters $\boldsymbol{A}^{(3)}, \boldsymbol{B}^{(3)}, \boldsymbol{C}^{(3)}$ from scratch. We note that this model is essentially identical to the baseline ConvNeXt-S3D except that each component of the factored kernels use standard 1D S4 layers instead of 1D local convolutions. Finally, by varying the initialization of these parameters, we can investigate additional factors affecting model training; in particular, we also run an ablation on the kernel timescales $\Delta$.

**Training.** Our training procedure is minimal, using only RGB frames (no optical flow). We sample clips of 2 seconds with 30 total frames at $224 \times 224$, followed by applying RandAugment; we performed a small sweep of the RandAugment magnitude for each model. All models are trained with learning rate 0.0001 and weight decay 0.2. Additional details are included in Appendix B.2.

Table 2: (**HMDB-51 Activity Recognition with ImageNet-pretrained models.**) (*Left*) Top-1 accuracy with 2D to 3SD kernel inflation. (*Right*) Ablation of initial temporal kernel lengths, controlled by S4's $\Delta$ parameter.

| | PARAMS | FLOW | RGB |
|---|---|---|---|
| Inception-I3D | 25.0M | 61.9 | 49.8 |
| ConvNeXt-I3D | 28.5M | - | 58.1 |
| ConvNeXt-S3D | 27.9M | - | 58.6 |
| S4ND-ConvNeXt-3D | 31.4M | - | **62.1** |

| INIT. LENGTH | ACC |
|---|---|
| 20.0 | 53.74 |
| 4.0 | 58.33 |
| 2.0 | 60.30 |
| 1.0 | 62.07 |

**Results.** Results are presented in Table 2. Our baselines are much stronger than prior work in this setting, $8\%$ top-1 accuracy higher than the original I3D model in the RGB frames only setting, and confirming that separable kernels (S3D) perform at least as well as standard inflation ($+0.53\%$). S4ND-ConvNeXt-3D improves over the baseline ConvNeXt-I3D by $+4.0\%$ with no difference in models other than using a temporal S4 kernel. This even exceeds the performance of I3D when trained on optical flow.

Finally, we show how S4ND's parameters can control for factors such as the kernel length (Table 2). Note that our temporal kernels $K^3(t^3)$ are always full length (30 frames in this case), while standard convolution kernels are shorter temporally and require setting the width of each layer manually as a hyperparameter [68]. S4 layers have a parameter $\Delta$ that can be interpreted such that $\frac{1}{\Delta}$ is the expected length of the kernel. By simply adjusting this hyperparameter, S4ND can be essentially initialized with length-1 temporal kernels that can automatically learn to cover the whole temporal length if needed. We hypothesize that this contributes to S4ND's improved performance over baselines.

Table 3: (**Settings for continuous capabilities experiments.**) Datasets and resolutions used for continuous capabilities experiments, as well as the model backbones used are summarized.

| DATASET | CLASSES | RESOLUTION | | | BACKBONE |
|---|---|---|---|---|---|
| | | base | mid | low | |
| CIFAR-10 | 10 | $32 \times 32$ | $16 \times 16$ ($2\times$) | $8 \times 8$ ($4\times$) | Isotropic |
| Celeb-A | 40 multilabel | $160 \times 160$ | $128 \times 128$ ($1.25\times$) | $64 \times 64$ ($2.50\times$) | ConvNeXt |

Table 4: **Zero-Shot Resolution Change.** Results for models trained on one resolution (one of low / mid / base), and zero-shot tested on another. Results are averaged over 2 random seeds.

| RESOLUTION | | CIFAR-10 | | | CELEB-A | |
|---|---|---|---|---|---|---|
| TRAIN | TEST | S4ND | CONV2D | FLEXNET-16 | S4ND | CONV2D |
| base | base | $\mathbf{93.10 \pm 0.22}$ | $91.9 \pm 0.2$ | $92.2 \pm 0.1$ | $\mathbf{91.75 \pm 0.00}$ | $91.44 \pm 0.03$ |
| mid | mid | $\mathbf{88.80 \pm 0.12}$ | $87.2 \pm 0.1$ | $86.5 \pm 2.0$ | $\mathbf{91.63 \pm 0.04}$ | $91.09 \pm 0.08$ |
| mid | base | $\mathbf{88.77 \pm 0.03}$ | $73.1 \pm 0.3$ | $82.7 \pm 2.0$ | $\mathbf{90.14 \pm 0.38}$ | $80.52 \pm 0.08$ |
| low | low | $\mathbf{78.17 \pm 0.13}$ | $76.0 \pm 0.2$ | - | $\mathbf{90.95 \pm 0.02}$ | $90.37 \pm 0.04$ |
| low | mid | $\mathbf{78.86 \pm 0.22}$ | $57.4 \pm 0.3$ | - | $\mathbf{84.44 \pm 1.04}$ | $80.45 \pm 0.11$ |
| low | base | $\mathbf{73.71 \pm 0.47}$ | $33.1 \pm 1.3$ | - | $\mathbf{84.73 \pm 0.54}$ | $80.59 \pm 0.14$ |

## 5.3 Continuous-signal Capabilities for Images

Images are often collected at varied resolutions, even with the same hardware, so it is desirable that models generalize to data sampled at different resolutions. We show that S4ND inherits this capability as a continuous-signal model, with strong zero-shot performance when changing resolutions, and the ability to train with progressively resized multi-resolution data. We perform an ablation to show that our proposed bandlimiting modification is critical to achieving strong performance when changing resolutions.

**Setup.** We focus on image classification on 2D benchmark datasets: a dataset with low-resolution images (CIFAR-10) and one with higher-resolution images (Celeb-A). For each dataset, we specify a base image resolution (base), and two lower resolutions (mid, low), summarized in Table 3. To highlight that S4ND's continuous capabilities are independent of backbone, we experiment with 2 different 2D backbones: an isotropic, fixed-width model backbone on CIFAR-10 [36] and a small ConvNeXt backbone on Celeb-A [43]. For each backbone, we compare S4ND's performance to Conv2D layers as a standard, widely used baseline. Additional details can be found in Appendix B.3.

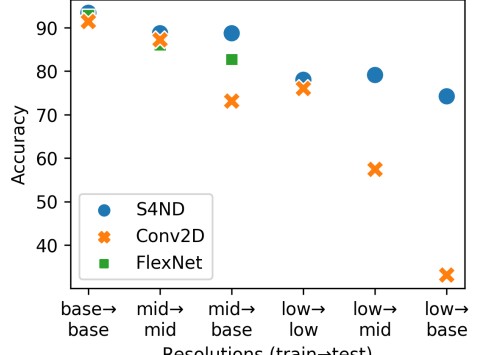

CIFAR-10: Zero-Shot Test Resolution Performance

Figure 3: (**CIFAR-10 zero-shot comparison.**) When trained and tested on the same resolutions, all models have similar performance (with S4ND slightly better). But, when trained and tested on different resolutions (the zero-shot setting), S4ND significantly outperforms Conv2D and FlexNet.

We first verify that S4ND achieves comparable test classification performance with baseline Conv2D models. Table 4 and Fig. 3 show that we exceed the performance of Conv2D on both tasks, with S4ND improving over Conv2D models by $0.4\%$ on CIFAR-10 and $0.3\%$ on Celeb-A.

**Zero-Shot Resolution Change.** We train S4ND and Conv2D models at either low or mid resolution, and test them at the base resolution for each dataset. We also compare to FlexConv [50] on CIFAR-10, which is the current SotA for zero-shot resolution change. Compared to training and testing

at the base resolution, we expect that Conv2D should degrade more strongly than S4ND, since it cannot adapt its kernel appropriately to the changed resolution. Table 4 and Fig. 3 show that S4ND outperforms Conv2D on mid → base by 15+ points and low → base by 40+ points on CIFAR-10, and 9+ points and 3+ points on Celeb-A. In fact, S4ND yields better performance on low → base (a more difficult task) than Conv2D does from mid → base (an easier task), improving by 1.1% on CIFAR-10 and 3% on Celeb-A. Compared to FlexConv on CIFAR-10 mid → base, S4ND improves zero-shot performance by 5+ points, setting a new SotA.

**Progressively Resized Training.** We provide an exploration of training with progressive resizing [14, 60] i.e. training in multiple stages at different resolutions. The only change we make from standard training is to reset the learning rate scheduler at the beginning of each stage (details in Appendix B.3). We compare S4ND and the Conv2D baseline with progressive resizing in Table 5.

For CIFAR-10, we train with a low → base, $80 - 20$ epoch schedule, and perform within $\sim 1\%$ of an S4ND model trained with base resolution data while speeding up training by 21.8%. We note that Conv2D attains much higher speedups as a consequence of highly optimized implementations, which we discuss in more detail in Section 6. For Celeb-A, we explore flexibly combining the benefits of both progressive resizing and zero-shot testing, training with a low → mid, $16 - 4$ epoch schedule that uses no base data. We outperform Conv2D by 7.5%+, and attain large speedups of 50%+ over training at the base resolution.

Table 5: (**Progressive resizing results.**) Validation performance for progressively resized training at base resolution, and speedup compared to training at base resolution on CIFAR-10 and Celeb-A. We use a $80 - 20$ and $16 - 4$ schedule for CIFAR-10 and Celeb-A, and also report performance training only at base resolution.

| DATASET | MODEL | EPOCH SCHEDULE | TRAIN RESOLUTION | VAL @ BASE RES. | SPEEDUP (STEP TIME) |
|---|---|---|---|---|---|
| CIFAR-10 | Conv2D | - | base | 91.90% | 0% |
| | S4ND | - | base | **93.40%** | 0% |
| | Conv2D | $80 - 20$ | low→ base | 90.94% | 51.7% |
| | S4ND | $80 - 20$ | low→ base | **92.32%** | 21.8% |
| CelebA | Conv2D | - | base | 91.44% | 0% |
| | S4ND | - | base | **91.75%** | 0% |
| | Conv2D | $16 - 4$ | low→ mid | 80.89% | 76.7% |
| | S4ND | $16 - 4$ | low→ mid | **88.57%** | 57.3% |

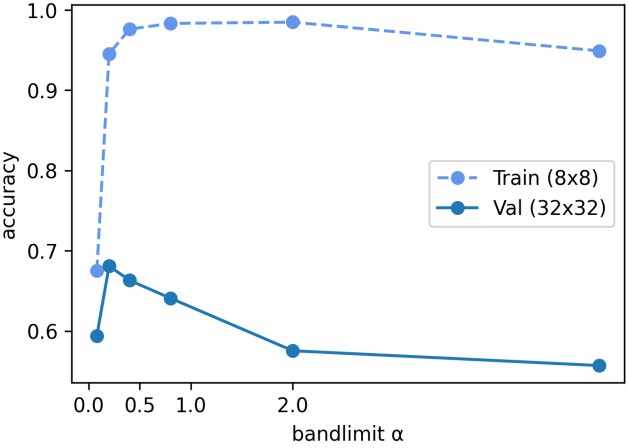
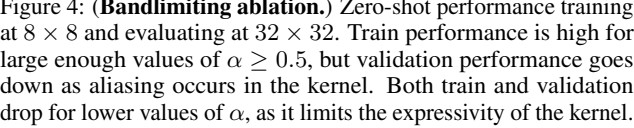

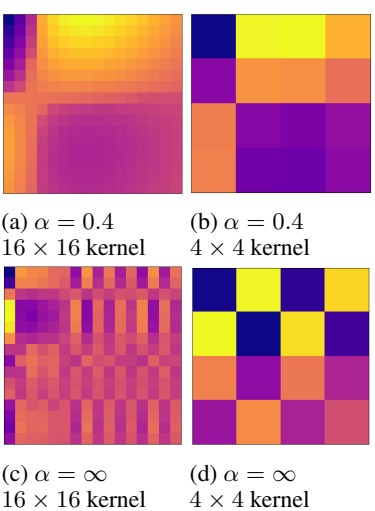

(a) $\alpha = 0.4$
$16 \times 16$ kernel

(b) $\alpha = 0.4$
$4 \times 4$ kernel

(c) $\alpha = \infty$
$16 \times 16$ kernel

(d) $\alpha = \infty$
$4 \times 4$ kernel

Figure 4: (**Bandlimiting ablation.**) Zero-shot performance training at $8 \times 8$ and evaluating at $32 \times 32$. Train performance is high for large enough values of $\alpha \geq 0.5$, but validation performance goes down as aliasing occurs in the kernel. Both train and validation drop for lower values of $\alpha$, as it limits the expressivity of the kernel.

Figure 5: (**Effect of bandlimiting on learned kernels.**) Bandlimiting significantly increases the smoothness of the kernels when resizing resolutions.

**Effect of Bandlimiting.** Bandlimiting in S4ND is critical to generalization at different resolutions. We analyze the effect of the bandlimiting parameter $\alpha$ on CIFAR-10 performance when doing zero-

shot resolution change. We additionally vary the choice of basis function $K_n(t)$ used in the SSM. Fig. 4 shows that zero-shot performance on base degrades for larger values of $\alpha$, i.e. for cutoffs that do not remove high frequencies that violate the Nyquist cutoff. As we would expect, this holds regardless of the choice of basis function. In Fig. 5, we visualize learned kernels with and without bandlimiting, showing that bandlimiting improves smoothness. Appendix B.3 includes additional experiments that analyze $\alpha$ on Celeb-A, and when doing progressively resized training.

# 6 Discussion

**Summary.** We introduced S4ND, a multidimensional extension of S4 that models visual data as continuous valued signals. S4ND is the first continuous model that matches SotA baselines on large-scale 1D and 2D image classification on ImageNet-1k, as well as outperforming a strong pretrained model in a 3D video classification setting. As a continuous-signal model, S4ND inherits useful properties that are absent from standard visual modeling approaches, such as zero-shot testing on unseen resolutions without a significant performance drop.

**Limitations.** A limitation of S4ND is its training speed in high dimensions. In the 1D image setting, S4ND-ViT has similar training speed to ViT; however, in the 2D setting, the S4ND-ConvNeXt was $2\times$ slower than the baseline ConvNeXt. We remark that vanilla local convolutions have been heavily optimized for years, and we expect that layers such as S4ND can be substantially sped up with more optimized implementations. Our core computational primitives accounting for $65\%$ of our runtime (FFT, pointwise operations, inverse FFT) are all bottlenecked by reading from and writing to GPU memory [57]. With a more optimized implementation that fuses these operations [9] (i.e., loading the input once from GPU memory, perform all operations, then write result back to GPU memory), we expect to speed these up by 2-3$\times$. Further discussion can be found in Appendix C.

**Future work.** We presented a first step in using continuous-signal models in images and videos, and believe this opens the door to new capabilities and directions. For example, recent video benchmark datasets are significantly larger than the HMDB-51 dataset used in our experiments [34, 18, 8, 46, 55], and can be explored. In addition, we demonstrated capabilities in zero-shot resolution *spatially*, but even less work has been done on zero-shot testing *temporally*. This opens up an exciting new direction of work that could allow models to be agnostic to different video sampling rates as well, capable of testing on higher unseen sampling rates, or irregular (non-uniform) sampling rates. As models become larger and combine multiple modalities, S4ND shows strong promise in being able to better model underlying continuous-signals and create new capabilities across visual data, audio, time-series (e.g., wearable sensors, health data), and beyond.

**Acknowledgments**

We thank Arjun Desai, Gautam Machiraju, Khaled Saab and Vishnu Sarukkai for helpful feedback on earlier drafts. This work was done with the support from the HAI-Google Cloud Credits Grant Program. We gratefully acknowledge the support of NIH under No. U54EB020405 (Mobilize), NSF under Nos. CCF1763315 (Beyond Sparsity), CCF1563078 (Volume to Velocity), and 1937301 (RTML); ONR under No. N000141712266 (Unifying Weak Supervision); ONR N00014-20-1-2480: Understanding and Applying Non-Euclidean Geometry in Machine Learning; N000142012275 (NEP-TUNE); the Moore Foundation, NXP, Xilinx, LETI-CEA, Intel, IBM, Microsoft, NEC, Toshiba, TSMC, ARM, Hitachi, BASF, Accenture, Ericsson, Qualcomm, Analog Devices, the Okawa Foundation, American Family Insurance, Google Cloud, Salesforce, Total, the HAI-AWS Cloud Credits for Research program, the Stanford Data Science Initiative (SDSI), and members of the Stanford DAWN project: Facebook, Google, and VMWare. The Mobilize Center is a Biomedical Technology Resource Center, funded by the NIH National Institute of Biomedical Imaging and Bioengineering through Grant P41EB027060. The U.S. Government is authorized to reproduce and distribute reprints for Governmental purposes notwithstanding any copyright notation thereon. Any opinions, findings, and conclusions or recommendations expressed in this material are those of the authors and do not necessarily reflect the views, policies, or endorsements, either expressed or implied, of NIH, ONR, or the U.S. Government.

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
