# A   Theory Details

We formalize and generalize the notation of Section 4.1 and prove the results.

For the remainder of this section, we fix a dimension $D$ (e.g. $D = 2$ for a 2-dimensional SSM). The $D$-dimensional SSM will be a map from a function $u : \mathbb{R}^D \to \mathbb{C}$ to $y : \mathbb{R}^D \to \mathbb{C}$.

**Definition 2** (Indexing notation). *Let $[d]$ denote the set $\{1, 2, \ldots, d\}$. Let $[0]$ denote the empty set $\{\}$. Given a subset $I \subseteq [D]$, let $-I$ denote its complement $[D] \setminus I$. Let $[-d] = -[d] = \{d + 1, \ldots, D\}$.*

**Definition 3** (Tensor products and contractions). *Given $a \in \mathbb{C}^{N_1}$ and $b \in \mathbb{C}^{N_2}$, let $a \otimes b \in \mathbb{C}^{N_1 \times N_2}$ be defined as $(a \otimes b)_{n_1, n_2} = a_{n_1} b_{n_2}$.*

*Given a tensor $x \in \mathbb{C}^{N_1 \times \cdots \times N_D}$ and matrix $\boldsymbol{A} \in \mathbb{C}^{N_1 \times N_1}$, define $\boldsymbol{A} \cdot^{(1)} x \in \mathbb{C}^{N_1 \times \cdots \times N_D}$ as*

$$(\boldsymbol{A} \cdot^{(1)} x)_{n_1, \ldots, n_D} = \sum_m \boldsymbol{A}_{n_1 m} x_{k, n_2, \ldots, n_D}.$$

*which is simply a matrix multiplication over the first dimension. Let $\cdot^{(\tau)}$ similarly denote matrix multiplication over any other axis.*

It will be easier to work directly from the convolutional definition of SSM (equation (5)). We will then show equivalence to a PDE formulation (equation (4)).

**Definition 4** (Multidimensional SSM). *Let $N_1, \ldots, N_D$ be state sizes for each of the $D$ dimensions. The $D$-dimensional SSM has parameters $\boldsymbol{A}^{(\tau)} \in \mathbb{C}^{N^{(\tau)} \times N^{(\tau)}}$, $\boldsymbol{B}^{(\tau)} \in \mathbb{C}^{N^{(\tau)}}$, and $\boldsymbol{C} \in \mathbb{C}^{N^{(1)} \times \cdots \times N^{(D)}}$ for each dimension $\tau \in [D]$.*

*and is defined by the map $u \mapsto y$ given by*

$$x(t) = (u * \bigotimes_{\tau=1}^{D} e^{t^{(\tau)} \boldsymbol{A}^{(\tau)}} \boldsymbol{B}^{(\tau)})(t)$$
$$y(t) = \langle \boldsymbol{C}, x(t) \rangle.$$

*Note that at all times $t = (t^{(1)}, \ldots, t^{(D)})$, the dimension of the state is $x(t) \in \mathbb{C}^{N_1 \times \cdots \times N_D}$.*

Finally, it will be convenient for us to define "partial bindings" of $u, x, y$ where one or more of the coordinates are fixed.

**Definition 5** (Partial binding of $u$). *Let $I \subseteq [D]$ and let $t^I = (t^{(I_1)}, \ldots, t^{(I_{|I|})})$ be an partial assignment of the time variables. Define $u_I(t^I)$ to be the function $\mathbb{R}^{D-|I|} \mapsto \mathbb{C}$ where the $I$ indices are fixed to $t^I$.*

*For example, if $D = 3$ and $I = [1]$, then*

$$u_I(t^I) = u(t^{(1)}, \cdot, \cdot)$$

*is a function from $\mathbb{R}^2 \to \mathbb{C}$ mapping $(t^{(2)}, t^{(3)}) \mapsto u(t^{(1)}, t^{(2)}, t^{(3)})$.*

**Definition 6** (Partial binding of $x$).

$$x_I(t^I)(t^{-I}) = (u_I(t^I) * \bigotimes_{\tau \in -I} e^{t^{(\tau)} \boldsymbol{A}^{(\tau)}} \boldsymbol{B}^{(\tau)})(t^{-I})$$

*Note that $x_{[0]}() : \mathbb{R}^D \mapsto \mathbb{C}^{N_1 \times \cdots \times N_D}$ coincides with the full state $x$.*

The following more formal theorem shows the equivalence of this convolutional LTI system with a standard 1D SSM differential equation in each dimension.

**Theorem 2.** *Given a partial state $x_I(t^I)$ and a time variable $t^{(\tau)}$ for $\tau \notin I$, let $I' = I \cup \{\tau\}$. Then the partial derivatives satisfy*

$$\frac{\partial x_I(t^I)}{\partial t^{(\tau)}}(t^{-I}) = \boldsymbol{A}^{(\tau)} \cdot^{(\tau)} x_I(t^I)(t^{-I})$$
$$+ \boldsymbol{B}^{(\tau)} \otimes x_{I'}(t^{I'})(t^{-I'})$$

*Proof.* WLOG assume $I = [d]$, since all notions are permutation-independent. We will consider differentiating the state $x_I$ with respect to the time variable $t^{(d+1)}$. The key fact is that differentiating a convolution $\frac{d}{dt}(f * g)$ is equivalent to differentiating one of the operands $f * (\frac{d}{dt}g)$.

$$
\begin{aligned}
\frac{\partial x_{[d]}(t^{[d]})}{\partial t^{(d+1)}}(t^{-[d]}) &= \frac{\partial}{\partial t^{(d+1)}}(u_I(t^I) * \bigotimes_{\tau \in -I} e^{t^{(\tau)}\boldsymbol{A}^{(\tau)}}\boldsymbol{B}^{(\tau)})(t^{-I}) \\
&= (u_I(t^I) * \frac{\partial}{\partial t^{(d+1)}} \bigotimes_{\tau \in -[d]} e^{t^{(\tau)}\boldsymbol{A}^{(\tau)}}\boldsymbol{B}^{(\tau)})(t^{-I}) \\
&= \left(u_I(t^I) * \left(\boldsymbol{A}^{(d+1)}e^{t^{(d+1)}\boldsymbol{A}^{(d+1)}}\boldsymbol{B}^{(d+1)} + \boldsymbol{B}^{(d+1)}\delta(t^{(d+1)})\right) \bigotimes_{\tau \in -[d+1]} e^{t^{(\tau)}\boldsymbol{A}^{(\tau)}}\boldsymbol{B}^{(\tau)}\right)(t^{-I}) \\
&= \boldsymbol{A}^{(d+1)} \cdot^{(d+1)} \left(u_I(t^I) * \bigotimes_{\tau \in -[d]} e^{t^{(\tau)}\boldsymbol{A}^{(\tau)}}\boldsymbol{B}^{(\tau)}\right)(t^{-I}) \\
&\quad + u_{[d+1]}(t^{[d+1]})\boldsymbol{B}^{(d+1)} \bigotimes_{\tau \in -[d+1]} e^{t^{(\tau)}\boldsymbol{A}^{(\tau)}}\boldsymbol{B}^{(d+1)} \\
&= \boldsymbol{A}^{(d+1)} \cdot^{(d+1)} x_{[d]}(t^{[d]})(t^{-[d]}) \\
&\quad + \boldsymbol{B}^{(d+1)} \otimes x_{[d+1]}(t^{[d+1]})(t^{-[d+1]})
\end{aligned}
$$

$\square$

The following corollary follows immediately from linearity of the convolution operator, allowing the order of convolution and inner product by $\boldsymbol{C}$ to be switched.

**Corollary A.1.** *The output $y$ is equivalent to $y = K * u$ where*

$$
K(t) = \langle \boldsymbol{C}, \bigotimes_{\tau=1}^{D} e^{t^{(\tau)}\boldsymbol{A}^{(\tau)}}\boldsymbol{B}^{(\tau)} \rangle
$$

This completes the proof of Theorem 1.

Finally, Corollary 4.1 is an immediate consequence of the Kronecker mixed-product identity

$$
(A \otimes B)(C \otimes D) = (AC) \otimes (BD).
$$

# B  Experimental Details

We use PyTorch for all experiments, and build on the publicly available S4 code.

## B.1  Image Classification

ImageNet training: all models were trained from scratch with no outside data using 8 Nvidia A100 GPUs. For both ViT and ConvNeXt experiments, we follow the procedure from T2T-ViT [69] and the original ConvNeXt [42], respectively, with minor adjustments. Preprocessing and dataloading was done using the TIMM [66] library. In S4ND-ViT, we turn off weight decay and remove the class token prepending of the input sequence. For the ConvNeXt models, we add RepeatAug [29], as well as reduce the batch size to 3840 for S4ND-ConvNeXt.

S4ND specific settings include a bidirectional S4 kernel followed by Goel et al [17], and a state dimension of 64 for the SSMs.

## B.2  Video Classification

HMDB-51 training: all models were trained from scratch on a single Nvidia A100 GPU. We use the Pytorchvideo library for data loading and minimal preprocessing of RGB frames only (no optical

Table 6: (**Performance on image classification.**) ImageNet settings for ViT and ConvNeXt baseline models.

|  | VIT | CONVNEXT |
|---|---|---|
| image size | $224^2$ | $224^2$ |
| optimizer | AdamW | AdamW |
| optimizer momentum | $\beta_1, \beta_2 = 0.9, 0.999$ | $\beta_1, \beta_2 = 0.9, 0.999$ |
| weight init | trunc. normal (std=0.02) | trunc. normal (std=0.02) |
| base learning rate | 0.001 | 0.004 |
| weight decay | 0.05 | 0.05 |
| dropout | None | None |
| batch size | 4096 | 4096 |
| training epochs | 300 | 300 |
| learning rate schedule | cosine decay | cosine decay |
| warmup epochs | 10 | 20 |
| warmup schedule | linear | linear |
| layer-wise lr decay [3, 6] | None | None |
| randaugment [7] | (9,0.5,layers=2) | (9,0.5,layers=2) |
| mixup [72] | 0.8 | 0.8 |
| cutmix [70] | 1.0 | 1.0 |
| repeataug [29] | None | 3 |
| random erasing [73] | 0.25 | 0.25 |
| label smoothing [59] | 0.1 | 0.1 |
| stochastic depth [30] | 0.1 | 0.1 |
| layer scale [63] | None | 1e-6 |
| head init scale [63] | None | None |
| exp.mov. avg (EMA) [48] | 0.9999 | None |

flow). During training, we randomly sample 2 second clips from each video. At validation and test time, 2 second clips are sampled uniformly to ensure that an entire video is seen. For each ConvNeXt 3D video model (baselines and S4ND versions), we fix the hyperparameters in Table 7, and do a sweep over the RandAugment[7] settings num_layers= $\{1, 2\}$ and magnitude= $\{3, 5, 7, 9\}$.

S4ND specific settings were similar to image classification, a bidirectional S4 kernel and a state dimension of 64 for the SSMs.

Table 7: (**Performance on video classification.**) HMDB-51 settings for all 3D ConvNeXt models (baselines and S4ND models).

|  | CONVNEXT 3D |
|---|---|
| image size | $224^2$ |
| # frames | 30 |
| clip duration (sec) | 2 |
| optimizer | AdamW |
| optimizer momentum | $\beta_1, \beta_2 = 0.9, 0.999$ |
| weight init | trunc. normal (std=0.02) |
| base learning rate | 0.0001 |
| weight decay | 0.2 |
| dropout (head) | 0.2 |
| batch size | 64 |
| training epochs | 50 |
| learning rate schedule | cosine decay |
| warmup epochs | 0 |
| stochastic depth [30] | 0.2 |
| layer scale [63] | 1e-6 |
| head init scale [63] | None |
| exp.mov. avg (EMA) [48] | None |

## B.3 Continuous Time Capabilities Experiments

We use the CIFAR-10 and Celeb-A datasets for all our experiments. Below, we include all experimental details including data processing, model training and hyperparameters, and evaluation details.

As noted in Table 3, we consider 3 resolutions for each dataset, a $\mathrm{base}$ resolution that is considered the standard resolution for that dataset, as well as two lower resolutions $\mathrm{low}$ and $\mathrm{mid}$. Images at the lower resolutions are generated by taking an image at the $\mathrm{base}$ resolution, and then downsampling using the `resize` function in `torchvision`, with bilinear interpolation and antialiasing turned on.

**CIFAR-10.** The base resolution is chosen to be $32 \times 32$, which is the resolution at which models are generally trained on this dataset. The lower resolutions are $16 \times 16$ and $8 \times 8$. We use no data augmentation for either zero-shot resolution change or progressive resizing experiments. We train with standard cross-entropy loss on the task of 10-way classification. For reporting results, we use standard accuracy over the 10 classes.

We use a simple isotropic model backbone, identical to the one used by Gu et al. [21] for their (1D) S4 model. The only difference is that we use either the S4ND or Conv2D layer in place of the S4 block, which ensures that the model can accept a batch of 2D spatial inputs with channels. For results reported in the main body, we use a $6 \times 256$ architecture consisting of 6 layers and a model dimension of 256.

For all CIFAR-10 experiments, we train for 100 epochs, use a base learning rate of 0.01 and a weight decay of 0.03.

**Celeb-A.** On Celeb-A, we consider $160 \times 160$ as the base resolution, and $128 \times 128$ and $64 \times 64$ as the lower resolutions. Images on Celeb-A are generally $218 \times 178$, and we run a $\mathrm{CenterCrop}(178) \rightarrow \mathrm{Resize}(160)$ transform in `torchvision` to resize the image. We use no data augmentation for either zero-shot resolution change or progressive resizing experiments. We train with standard binary cross-entropy loss on the task of 40-way multilabel attribute classification on Celeb-A. For reporting results, we use the multilabel binary accuracy, i.e. the average binary accuracy across all 40 tasks.

Similar to our ImageNet experiments, we use a ConvNeXt model as the basic backbone for experiments on Celeb-A. However, use a particularly small model here, consisting of 4 stages with depths $(3, 3, 3, 3)$ and corresponding model dimensions $(64, 128, 256, 512)$. For S4ND, we simply follow the same process as ImageNet, replacing all depthwise Conv2D layers with S4ND. A minor difference is that we use a global convolution in the stem downsampling for S4ND, rather than the smaller kernel size that we use for ImageNet. We use a drop path rate of 0.1 for both S4ND and the Conv2D baseline in all experiments.

For all Celeb-A experiments, we train for 20 epochs, use a base learning rate of 0.004, and use automatic mixed precision for training.

**Other fixed hyperparameters.** For all experiments, we use bidirectional S4 kernels following Goel et al. [17], and use a state dimension of 64 for the S4 SSMs. We use AdamW as the optimizer.

### B.3.1 Zero-Shot Resolution Change

For Table 4, we simply train on a single resolution among $\mathrm{base}$, $\mathrm{mid}$ and $\mathrm{low}$, and then directly test at all higher resolutions. For model selection at a particular test resolution, we select the best performing model for that test resolution i.e. we checkpoint the model at the epoch where it performs best at the test resolution in question. Note that we refer to "test resolutions" but report validation metrics in Table 4, as is standard practice for additional experiments and ablations.

**CIFAR-10.** We use a batch size of 50, and for both methods (and all resolutions), we use a cosine decay schedule for the learning rate with no restarts and a length of 100000, and a linear warmup of 500 steps.

For baseline Conv2D hyperparameters, we compare the performance of Conv2D with and without depthwise convolutions. Otherwise, all hyperparameters are fixed to the common values laid out in the previous section (and are identical to those used for S4ND).

For S4ND hyperparameters, we sweep the choice of initialization for the state space parameters $A$, $B$ in the S4 model among {legs, fourier, random-linear, random-inv}[2] and sweep bandlimit ($\alpha$) values among $\{0.05, 0.10, 0.20, 0.50, \infty\}$.

**Celeb-A.** We use a batch size of 256 when training on the lower resolutions, and 128 when training at the base resolution. For both methods (and all resolutions), we use a cosine decay schedule for the learning rate with no restarts and a length of 13000 (batch size 256) or 26000 (batch size 128), and a linear warmup of 500 steps.

For baseline ConvNeXt hyperparameters, we use a weight decay of 0.1 after an initial sweep over weight decay values $\{0.05, 0.1, 0.2, 0.5\}$.

For S4ND hyperparameters, we sweep the choice of initialization for the state space parameters $A$, $B$ in the S4 model among {legs, fourier, random-linear, random-inv}, sweep bandlimit ($\alpha$) values among $\{0.05, 0.10, 0.20, 0.50, \infty\}$, and use a weight decay of 1.0. The value of the weight decay was chosen based on an initial exploratory sweep over weight decay values $\{0.1, 0.2, 0.5, 1.0, 5.0\}$.

### B.3.2 Progressive Resizing

For Table 5, we train with a resizing schedule that progressively increases the resolution of the data. We report all metrics on validation sets, similar to our zero-shot experiments. We only use 2 resizing stages in our experiments, as we found in preliminary experiments that 2+ stages had little to no benefit.

All hyperparameters and training details are identical to the zero-shot resolution change experiments, except for those we describe next. When training with multiple stages, we reset the cosine learning rate scheduler at the beginning of each stage (along with the linear warmup). The length of the scheduler is varied in proportion to the length of the stage, e.g. on CIFAR-10, for a stage 50 epochs long, we change the length of the decay schedule from 100000 steps to 50000 steps (and similarly for Celeb-A). We always use a linear warmup of 500 steps for each stage. Additionally, Table 5 denotes the length (in epochs) of each stage, as well as their training resolutions.

Another important detail is that we set the bandlimit parameter carefully for each stage. We found that bandlimiting in the first stage is critical to final performance, and without bandlimiting the performance on the base resolution is substantially worse. Bandlimiting is generally not useful for a stage if it is training at the base resolution i.e. at the resolution that we will be testing at (in our experiments, this is the case only for second stage training on CIFAR-10).

On CIFAR-10, we use a bandlimit $\alpha$ of 0.10 or 0.20 for the first stage, depending on whether the first stage was training at $8 \times 8$ (low) or $16 \times 16$ (mid) respectively. In the second stage, we always train at the base resolution and use no bandlimiting, since as stated earlier, it has a negligible effect on performance.

For Celeb-A, we set the bandlimit parameter to 0.1 for both stages. Note that on Celeb-A, we do not train at the base resolution at all, only training at low in the first stage and mid in the second stage.

## C  Discussion

**Runtime profiling and potential optimization.**  We profile the runtime of an S4ND layer on an A100 GPU, and plot the time taken by each operation in Fig. 6. We see that the majority of the time (between 65% and 80%) are taken by the FFT, pointwise operation, and inverse FFT. The higher-dimensional FFT is a very standard scientific primitive that should be substantially optimizable.

These operations are memory-bound, that is, the runtime is dominated the time to read/write to GPU memory. There is currently no library support for fusing these steps, so for each of those steps the data has to be loaded from GPU memory, arithmetic operations are performed, then the result is written back to GPU memory. One potential optimization is *kernel fusion*: the input could be loaded once, all the arithmetic operations for all three steps are performed, then the final result is written back to GPU memory. We expect that with library support for such optimization, the S4ND runtime can be reduced by 2-3×.

---

[2]These correspond to initializations of the model that allow the kernel to be expressed as a combination of different types of basis functions. For more details, we refer the reader to Gu et al. [19, 21].

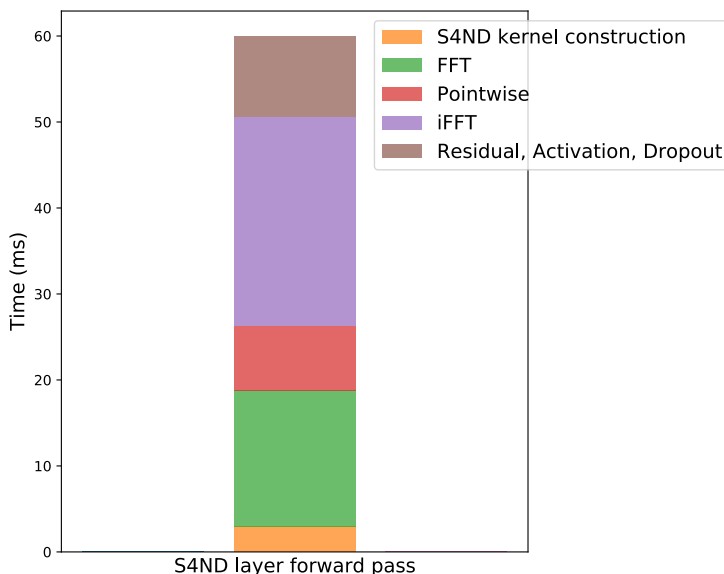

Figure 6: Timing breakdown of different steps in the S4ND forward pass, for a batch of 64 inputs, each of size $224 \times 224$, on an A100 GPU. FFT, pointwise operation, and inverse FFT take between 65% and 80% of the time.