# OpenReview forum: "S4ND: Modeling Images and Videos as Multidimensional Signals with State Spaces"
_NeurIPS.cc/2022/Conference — NeurIPS 2022 Accept_

### Official Review · Reviewer_G2NP · 2022-06-23

**Rating:** 7
**Confidence:** 3
**Soundness:** 4 excellent
**Presentation:** 4 excellent
**Contribution:** 3 good

**Summary:**

 This paper extends the state space model, S4, to multi-dimensional. As a result, the model can treat input as higher-dimensional continuous signal. The authors discussed resolution change and bandlimiting in relation to Nyquist cutoff frequency. Experiments are performed on state-of-the-art models on various vision tasks, such as image classification and video classification. The authors showed that S4ND performs well across 1D to 3D, while also showing the benefit of training on one resolution and testing on another.

POST-REBUTTAL UPDATE:

I have read the authors' rebuttal, and have decided to keep the original score of 7.

**Questions:**

I find Figure 4 (Effect of bandlimiting on learned kernels) to be really interesting. It really shows that without bandlimiting, the model can learn quite high-frequency kernels. My questions all revolve around this:

1. What was the process of choosing alpha on the image classification and video classification experiments, and what are the values chosen for the tables?
2. Have you tried analyzing the frequency distribution of trained kernels of CNNs / ViTs? Does that most resemble S4ND when the optimal alpha is chosen?
3. Nit: Should "Section 5.3" in L321 be "Figure 4"?

**Limitations:**

The authors had a paragraph discussing limitations, which I find to be candid.

**Strengths And Weaknesses:**

Strength:
- Extending the S4 model from 1D to high-dimension unleashed its use case significantly
- Experiments are thorough, and cover across 1D to 3D vision tasks
- The baselines used in experiments are recent and strong ones

Limitations:
- Some might feel that the generalization from 1D to high-dimension is quite straightforward
- Some questions on bandlimiting: see my other sections

Overall I think the originality may be medium, but quality, clarity, and significance are all high.

---

> ### Author Response · Authors · 2022-08-02
> **Response to Reviewer G2NP**
>
> We thank the reviewer for their time, and their feedback regarding the thoroughness of the experiments as well as thoughtful questions regarding the bandlimiting contribution.
>
> ## S4ND generalization
>
> In hindsight, the generalization of S4 from 1D to higher dimension has a simplicity and elegance to its formulation, in large part due to its empirical performance as a drop-in replacement on large-scale images and videos. However, this was not immediately obvious a priori. The original S4 worked only on low dimensional 1D sequences, and previous continuous models were only successful on toy image datasets like CIFAR-10, yet alone even attempted on video. We were able to outperform previous continuous models and the baseline discrete models.
>
> ## Clarifications on Bandlimiting
>
> We are glad the reviewer found the effect of bandlimiting interesting.  We hypothesized that higher frequencies needed to be regularized to fully exploit the resolution generalization abilities of S4ND, where we used a low-pass bandlimiting filter to control these frequencies. We performed a full sweep of the $\alpha$ hyperparameter (where lower values correspond to lower frequencies passed) to gather baselines and to visually compare the smoothness of the kernels. Figure 2 shows the effect of $\alpha$ as it is increased (allowing more high frequencies), and we use the argmax value of $\alpha$ corresponding to the top accuracy performance.
>
> We did not analyze the frequency distribution of trained weights in CNNs and ViT, which is an intriguing idea we leave open to future research.

---

### Official Review · Reviewer_wFo3 · 2022-07-11

**Rating:** 8
**Confidence:** 3
**Soundness:** 3 good
**Presentation:** 3 good
**Contribution:** 3 good

**Summary:**

This work build on a recent line of work on deep state-space models (SSMs). It proposes a new multidimensional SSM layer that extends SSMs’ continuous-signal modeling ability to multidimensional data including images and videos. The results show that the method can model large-scale visual data in 1D, 2D, and 3D as continuous multidimensional signals and demonstrate strong performance by simply swapping Conv2D and self-attention layers with S4ND layers in existing state-of-the-art models. The authors further experiment on imagenet-1K with recent state-of-the-art methods like ViT, ConvNext. This work is novelty, strong, and solid.

**Questions:**

This paper is well-written, organized, and clearly summarized, so I do not have any questions on it.

**Limitations:**

The authors summarized the limitations in the conclusion section well.

**Strengths And Weaknesses:**

The paper seems well written, the theoretical and experimental parts are interesting and relevant. The novel contribution which is to turn the standard SSM (a 1D ODE) into a multidimensional PDE governed by an independent SSM per dimension, the authors did a good job at summarizing the key ideas. The results show that the method can model large-scale visual data in 1D, 2D, and 3D as continuous multidimensional signals and demonstrate strong performance by simply swapping Conv2D and self-attention layers with S4ND layers in existing state-of-the-art models. The authors further experiment on imagenet-1K with recent state-of-the-art methods like ViT, ConvNext and achieve impressive performance.

---

> ### Author Response · Authors · 2022-08-02
> **Response to Review wFo3**
>
> We thank the reviewer for their time reviewing our work, and their comments highlighting the clarity of the paper and impressive performance of S4ND.

---

### Official Review · Reviewer_Epuh · 2022-07-11

**Rating:** 7
**Confidence:** 3
**Soundness:** 3 good
**Presentation:** 4 excellent
**Contribution:** 3 good

**Summary:**

This paper proposes a novel continuous signal model S4ND, which extends the prior work S4 to multidimensional signals and can serve as a drop-in replacement in modern vision architectures. The authors show improvement in image and video classification tasks. The proposed continuous signal model has several useful properties, including zero-shot generalization to unseen resolutions and progressive resized training.

**Questions:**

1. What are the number of parameters, FLOPs, and inference speed for different models in the main tables? Having this information would be helpful for better model comparison.
2. In Table 4, after training with low-resolution images, testing with high-resolution images has worse performance than testing with low-resolution images. Does this have any practical applications in real-world applications? For example, a baseline is to down-sample the test image to the training resolution, so that the performance would not drop too much.

**Limitations:**

Limitations and potential negative societal impacts are well addressed.

**Strengths And Weaknesses:**

### Strengths
+ The proposed method is well-motivated with theoretical derivation and is a natural generalization from the prior work.
+ The effectiveness of the proposed method is evaluated on common benchmarks with large-scale datasets. The comparison is in a fair setting and the baselines are solid.
+ The proposed S4ND can serve as a simple drop-in replacement in modern vision architectures, which is simple to implement and can have flexible applications.

### Weaknesses
- Despite the consistent improvement, the performance gap is not very large (especially for image classification, most of the improvements seem to be less than 1%), which makes it less practical.
- When comparing to the baseline models, the model size and efficiency seem to be unclear, from the main paper.

---

> ### Author Response · Authors · 2022-08-02
> **Response to Reviewer Epuh**
>
> We thank the reviewer for their thoughtful review and comments about the effectiveness of S4ND on large-scale benchmarks.
>
> ## Performance gap to baselines
>
> For ImageNet, state-of-the-art models being beaten by less than 0.1 points is not uncommon [3]. We note that the performance increase of +1.5 points for S4ND in ViT for ImageNet (just by replacing self-attention for S4ND layers) is quite significant in this highly competitive benchmark. For videos, S4ND was able to achieve a +3.5 point gain on the HMDB-51 dataset, another highly competitive video dataset. In addition, when comparing to previous state-of-the-art continuous models, S4ND outshines further, where ImageNet and videos were not even attempted (or published) but only toy datasets like CIFAR-10 were used [1, 2].
>
> ## Efficiency
>
> In regards to parameter count, the S4ND variants were designed to maintain approximately the same model size as their corresponding baselines (ViT/ConvNeXt) as described in section 5.1 on L217. We plan to add FLOPs comparison for S4ND vs. baselines.
>
> [3] https://paperswithcode.com/sota/image-classification-on-imagenet

---

### Official Review · Reviewer_gCjF · 2022-07-12

**Rating:** 6
**Confidence:** 3
**Soundness:** 3 good
**Presentation:** 3 good
**Contribution:** 2 fair

**Summary:**

This paper targets modeling high dimensional data/features with continuous signals -- State Space Models (SSMs) and proposes S4ND based on S4. The proposed model is able to work with different dimensional data including1D, 2D, and 3D. The authors validate the proposed model at different types of benchmarks and the experimental results indicate that the:

1) S4ND exceeds the performance of 1D-based ViT, and 2D-based ConvNeXt by replacing the original backbone with S4ND layers.

2) S4ND can overcome the aliasing issue of the original S4 by equipping a simple band-limiting modification, so that is able to generalize from low resolution to higher ones.

3) S4ND can also speed up the training for high-resolution models.

**Questions:**

See the previous block for the questions and concerns.

**Justification of rating:** Personally I love the SMM-based models and I think this work, specifically, is well-written, and the experiments are sufficient. Currently, I give a borderline rej mainly due to the novelty -- I think the S4ND model is more like a simple extension of S4 from 1D to 2D. For the 3D model, it is not a universal model, but a combination of the 1D S4 and the 2D S4. I am willing to upgrade my rating if the authors can address this major concern, otherwise, I would like to keep my rating.

---
Thanks to the authors for the feedback. After checking the other reviews and the response (re-checked eq 1 to 5), my major concern has been resolved. Though the novelty of straightforwardly extending S4 from 1D to higher dimensions might be limited, but I agree with the other reviewers that this paper provides a drop-in replacement for CNN and is easy to implement in modern learning architectures. I upgraded my rating after seeing the response.

**Limitations:**

Limitations have been discussed in the submission and there is no obvious negative social impact of this work.

**Strengths And Weaknesses:**

## Strengths

**Originality:** The S4ND is based on the novel and interesting SSM-based models. Compared with the classic conv-based models, SSMs can build continuous models for the input data. Compared with the previous SMMs, S4ND first extends the modeling strategy from 1D to higher dimensions.

**Quality:** The paper is overall well-written and the whole *high-level* architecture is easy to follow. The experimental evaluations are sufficient.

**Significance:** Experimental evaluations and ablations can mostly support what the authors claimed.

---
## Weaknesses

Though overall the paper is good, there are still some major concerns from my side:

1) **Presentation and novelty** is the biggest concern for me. The whole model is based on S4[17] and the annotations are derived partially from [16]. It takes some time to fully understand the overall idea. It's hard for me to evaluate the see if it is trivial to extend the S4 format from 1D to 2D based on eq (1)~(4) but it seems trivial to me. So novelty is one major concern.

2) **S4ND for 3D or higher dimensions:** In Sec. 5.2, the authors build the 3D model by operating a 1D S4 (temporal) on top of 2D S4, but not a pure 3D S4DN model. This model works for video-based data. However, another type of 3D data is the volume-based data, such as the CT or MRI data for medical image processing. In this case, the 3 x,y,z dimensions cannot be separated into two groups since they are equally weighted. Then, how does the proposed S4ND model generalize from 2D to 3D? In other words, I think the S4ND actually generalizes the S4 model from 1D to 2D, but not to higher/multiple dimensions. The video-based S4ND is a special case that combines the 1D model and the 2D model but overall the model is not a multi-dimensional model.

Some other minor concerns:

3) **The training time:** In L331-339, the authors mentioned that the training time of the S4ND-based models is slower than the baselines. Though in Fig. 4 in the appendix, the authors addressed that the FFT, iFFT and pointwise operations took a long time. But Fig.4 is for the forward pass for the S4ND module, not the whole model. I wonder if the FFT, iFFT, and pointwise operations also take a large weight of time consumption in terms of the full model? Besides, since no codes are attached, I take the codebase of [16] as a reference. The FFT and IFFT are implemented in pytorch, is it possible to make the whole model runnable with GPU? And if that's the case, a direct comparison of FLOPS between the baseline and S4DN can make more sense.

4) **Speedup results in Table 5:** The authors claimed that ```S4ND comes within ∼ 1% of a high-resolution model while training 22% faster```. This claim is correct but kind of misleading. In Table 5, compared with the conv-based baselines, S4ND-based models are still 1.5~2 times slower.

5) **Real-world applications:** The final minor concern is about the real-world application of the SSM-based models. Though I agree that these models are in general more powerful and generalizable. However, it is also more complicated and time-consuming. We are also not sure about the robustness of SSM-based models in comparison to the convolution-based models. Based on the experimental results, the performance gaps between the S4ND-based models and the conv-based models are not big. A more powerful backbone and structure may also probably make up this gap. Then, what's the motivation for building the SMM-based networks instead of the convolution-based ones? (No need to address this concern, it is just some thought about the applications of SMM-based models)

---
Typos:

1) Appendix section references are incorrect (Sec. D should be Sec. C, Sec. C.x should be Sec. B.x)

---

> ### Author Response · Authors · 2022-08-02
> **Response to Reviewer gCjF**
>
> We thank the reviewer for their detailed review of our work and are glad they appreciate the potential of SSM-based models. Below, we clarify the reviewer’s two primary concerns about S4ND’s ability to model arbitrary dimensions, and the novelty of our work. We then respond to additional clarifications that were requested.
>
> ## S4ND in arbitrary dimensions
>
> We clarify the confusion raised by the reviewer around S4ND’s ability to model higher dimension data.
>
> Indeed, **S4ND can be used to model data in an arbitrary number of dimensions**. Conceptually, S4ND can be viewed as an (arbitrary dimension) CNN with two changes to the convolution kernel (itself an N-dimensional tensor):
>
> 1. The kernel is factored as the outer product of N distinct 1-D kernels. More generally, the kernel can be a low-rank tensor.
> 2. Each 1-D kernel is parameterized as an S4 kernel.
>
> This view of S4ND (equation (5)) shows that *all dimensions are symmetric and S4ND can be interchanged anywhere that a vanilla (ND) CNN can be used*, including 3D MRI data.
> An alternate view of S4ND (equations (1)-(4)) shows that it can be represented as a partial differential equation, endowing it with continuous modeling capabilities.
>
> On video data, *S4ND can also be trained from scratch as a pure 3D model, where it matches a 3D CNN baseline*. Our video experiment used a “2D+1D” kernel in order to emulate the baseline CNN, which uses a popular technique of kernel inflation in video modeling (I3D [4]; S3D [5]) that outperforms training from scratch. *Because of the outer product formulation, this 2D+1D kernel is a special case of the general S4-3D kernel.*
>
> ## Novelty and Contributions
>
> As detailed above, S4ND is a method to reduce general ND convolutions into 1D kernels that can be parameterized using a powerful black box representation such as S4. *We view the simplicity of this model as a strength* that opens new directions for SSM-based models to be applied to more general data.
>
> Although the reviewer focused on methodological novelty, we emphasize that our work also presents important *empirical contributions*, showing that a principled continuous model for ND data can be **practical and performant on large-scale images and videos**.
>
> A detailed explanation of the contributions of our work is included in the common response to all reviewers.
>
> ## Additional Clarifications
>
> ### Training time
>
> We provide training time comparisons for the **whole model on Line 333** and per-operation breakdown of a **single S4ND layer in Figure 4** of the Appendix. We believe that a breakdown of the S4ND layer is more informative than the overall model, as this is the core difference from baselines, since every other component is held constant in our experiments. We note that outside of the S4ND layer, the only other operations in the network are a LayerNorm and residual connection which should be relatively negligible.
>
> ### Speedup results in Table 5
>
> The statement “S4ND comes within ∼1% of a high-resolution model while training 22% faster” is supported by Table 5 (CIFAR-10 S4ND results). L309-312 include a detailed discussion of this finding, with additional context that points out the higher speedups attained by Conv2D along with a forward reference to Section 6, where we candidly discuss the difference in speed for S4ND and ConvND (“Limitations”; L331-339).
>
> ### Real-world applications
>
> As with many novel approaches, there can be a learning curve to existing methods. It is our hope that as we (and others) continue to research SSMs, we can leverage their superior theoretical advantages to increase the performance gap, and reduce the computation constraints. We believe it’s still early, and there’s still a lot of runway and opportunity to increase the efficiency of SSMs as the community gets more familiar with the benefits of these models. For example, as FFTs and depthwise convolutions become more prevalent and optimized on modern hardware, this will improve S4ND significantly.
>
> **Citations**
>
> [4] Joao Carreira and Andrew Zisserman. Quo vadis, action recognition? a new model and the kinetics dataset. In proceedings of the IEEE Conference on Computer Vision and Pattern Recognition, pages 6299–6308, 2017
>
> [5] Saining Xie, Chen Sun, Jonathan Huang, Zhuowen Tu, and Kevin Murphy. Rethinking spatiotemporal feature learning for video understanding. arXiv preprint arXiv:1712.04851, 1(2):5, 2017.

---

### Author Response · Authors · 2022-08-02
**Response to All Reviewers**

We thank the reviewers for their thoughtful reviews and suggestions. We’re glad that the reviewers found our work to be well written and motivated, and that our baselines and experiments support the performance of S4ND. Below, we provide a shared response that reemphasizes the contributions of our work for all reviewers, and we respond individually to concerns raised by each reviewer as well.

### Summary of our work

The central contribution of our work is to develop a multidimensional model (S4ND) with continuous modeling properties that is competitive with state-of-the-art models on large-scale benchmarks like ImageNet. This represents a departure from prior work in continuous modeling, which previously lagged state-of-the-art models in performance and were confined to small-scale benchmarks due to performance and computational issues. [1, 2]

### Methodological novelty

The original S4 model is restricted to modeling sequences, and we introduced S4ND as a simple and principled *extension of S4 to model data of arbitrary dimensions* like images and videos. We emphasize that this extension is conceptually interesting since unlike standard ND convolutions, S4ND has continuous modeling capabilities, such as zero-shot generalization to new resolutions at test time, or the ability to take advantage of progressive resizing to speed up training. To realize these continuous capabilities in practice, we also introduce a *novel bandlimiting method for S4ND* that controls the frequencies learned by the S4ND kernel during training.

### Empirical contributions

An important contribution of this work is demonstrating that *continuous models can be competitive with standard state-of-the-art vision models*, while having additional capabilities like zero-shot resolution change. To the best of our knowledge, this has not previously been shown – no other continuous model is competitive with state-of-the-art models on large-scale benchmarks like ImageNet, or on applications such as video classification.

In particular, we apply S4ND to large-scale vision tasks in 1D, 2D and 3D as a drop-in replacement for self-attention (1D) or standard 2D or 3D convolutions, improving performance in all settings. Finally, we show that in addition to strong performance, S4ND’s continuous modeling allows it to better generalize across resolutions both compared to standard models and previously introduced continuous models, e.g. outperforming a baseline Conv2D by 40 points on zero-shot CIFAR-10 resolution change.

**Citations**

[1] David W. Romero, Robert-Jan Bruintjes, Jakub M. Tomczak, Erik J. Bekkers, Mark Hoogen doorn, and Jan C. van Gemert. “FlexConv: Continuous kernel convolutions with differentiable kernel sizes.” arXiv, abs/2110.08059, 2021.

[2] David W Romero, Anna Kuzina, Erik J Bekkers, Jakub M Tomczak, and Mark Hoogendoorn. “CkConv: Continuous kernel convolution for sequential data.” arXiv preprint arXiv:2102.02611, 2021.

---

### Meta-Review · Area_Chair_qppb · 2022-08-23

**Recommendation:** Accept
**Confidence:** Certain

**Metareview:**

The paper develops a multidimensional model (S4ND) with continuous modeling properties, which outperforms state-of-the-art models on large-scale benchmarks. All four reviewers reach an agreement and vote for accepting the paper. Also, the rebuttal and the discussion have successfully addressed all the major concerns raised by the reviewers. AC agrees with all reviewers’ judgments and recommends accepting the paper because of the novelty and high performance of the proposed model.

**Award:**

No

---

### Decision · Program_Chairs · 2022-09-14

Accept